



# Assessing the impact of acid rain and forest harvest intensity with the HD-MINTEQ model – Soil chemistry of three Swedish conifer sites from 1880 to 2080

Eric McGivney[1], Salim Belyazid[2], Therese Zetterberg[3], Stefan Löfgren[4], Jon Petter Gustafsson[1,5]

[1]Department of Sustainable Development. Environmental Science and Engineering, KTH Royal Institute of Technology, Teknikringen 10B, 100 44, Stockholm, Sweden.

[2]Department of Physical Geography, Stockholm University, SE-106 91 Stockholm, Sweden

[3]IVL Swedish Environmental Research Institute Ltd., P.O. Box 53021, SE-400 14 Göteborg, Sweden

[4]Department of Aquatic Sciences and Assessment, Swedish University of Agricultural Sciences, P.O. Box 7050, SE-750 07 Uppsala, Sweden

[5]Department of Soil and Environment, Swedish University of Agricultural Sciences, P.O. Box 7014, SE-750 07 Uppsala, Sweden.

Correspondence: Jon Petter Gustafsson (gustafjp@kth.se)

**Abstract**. Forest soils are susceptible to anthropogenic acidification. In the past, acid rain was a major contributor to soil
acidification, but now that atmospheric levels of S have dramatically declined, concern has shifted towards biomass-induced acidification, i.e., decreasing soil solution pH due to tree growth and harvesting events that permanently remove base cations (BC) from forest stands. We use a novel dynamic model, HD-MINTEQ, to investigate the long-term impacts of two theoretical future harvesting scenarios in the year 2020, a conventional harvest (CH, which removes stems only) and a whole-tree harvest (WTH, which removes 100% of the above-ground biomass except for stumps), on soil chemistry and weathering rates at three
different Swedish forest sites (Aneboda, Gårdsjön, and Kindla). Furthermore, acidification following the harvesting events is compared to the historical acidification that took place during the 20th century due to acid rain. Our results indicate that historical acidification due to acid rain had a larger impact on pore water chemistry and mineral weathering than tree growth and CH or WTH events, at least if nitrification remained at a low level. However, compared to a no-harvest scenario (NH), WTH and CH significantly impacted soil chemistry and weathering rates. Directly after a harvesting event (CH or WTH), the
soil solution pH sharply increased for 5 to 10 years before slowly declining over the remainder of the simulation (until year 2080). WTH acidified soils slightly more than CH, with the largest effects being seen for the B1 horizons by the year 2080.





Even though the pH values in the WTH and CH scenario decreased with time as compared to NH, they did not drop to the levels observed around the peak of historic acidification (1980-1990), indicating that the pH decrease due to tree growth and harvesting would be less impactful than that of historic atmospheric acidification. Weathering rates differed across locations and soil layers in response to historic acidification, but at several sites and layers, annual weathering rates decreased in tandem

with decreasing pH, which is likely due to $Al^{3+}$ weathering brakes. Weathering rates after the harvesting scenarios in 2020 generally increased although the dynamics were quite different depending on the site and soil layer.

## 1. Introduction

Anthropogenic acidification has an impact on soils, streams, organisms, agriculture, and forestry. The acidification of soils is influenced by both vegetation growth and atmospheric deposition. During the 20th century, sulfur (S) deposition, which peaked

in the 1980s, was the primary source of acidification in the acidic forest soils of the northern hemisphere (van Breemen et al., 1984). However, now that S deposition has dropped to around the 1930s level throughout Western Europe (Bertills et al., 2007; Engardt et al., 2017), focus has shifted towards understanding forest soil dynamics in response to forest biomass production and different harvesting scenarios (Akselsson et al., 2007; Iwald et al., 2013; de Jong et al., 2017).

Tree growth acidifies the soil through the net uptake of cations over anions, which results in an accumulation of $H^+$ in the form of organic acids (Nilsson et al., 1982). Forests that are recurrently harvested for lumber and paper production are especially susceptible to biomass-induced acidification, such as those in northern Europe. Mass balance calculations show considerable losses of base cations $Ca^{2+}$, $Mg^{2+}$, $Na^+$, and $K^+$ (BC) due to forest management practices, which may have strong acidifying effects on soils of base-poor mineralogy (Akselsson et al., 2007; Iwald et al., 2013). Therefore, there is a need to develop

sustainable forestry practices in which the net losses of BC are minimized to avoid acidification and long-term depletion of BC (Vadeboncoeur et al., 2014).

Models that can accurately predict forest soil chemistry based on uptake trends, plant growth, mineral weathering, harvesting scenarios, and deposition rates are powerful in assessing the susceptibility of soils to biomass-induced acidification. MAGIC

(Cosby et al., 1985, 2001) and ForSAFE (Wallman et al., 2005) are two such models that have been frequently used for this purpose, c.f. Gustafsson et al. (2018). Due to historic data collection and well-documented forestry practices, Swedish forests provide an excellent setting to develop and validate such models. For example, Belyazid et al. (2006) used ForSAFE to simulate changes in soil chemistry relative to atmospheric deposition at 16 different forest sites across Sweden, and showed that enhanced tree growth due to elevated nitrogen deposition could delay or even reverse the recovery of soils from acidification

caused by the historical acid deposition. Zetterberg et al. (2014) used MAGIC to simulate changes in soil $Ca^{2+}$ pools and stream acid neutralizing capacity at multiple harvest scenarios for three different Swedish forest stands. Large depletions of soil exchangeable $Ca^{2+}$ in response to whole-tree harvesting (WTH) were predicted by the model when compared to conventional



harvesting (CH). However in a complementary study, based on data from three Swedish experimental sites with stem-only and WTH treatments, it was found that MAGIC exaggerated the $Ca^{2+}$ loss due to harvesting between 1990 and 2013 (Zetterberg et al., 2016).

In this paper, we describe the soil chemical dynamics of three different Swedish forest stands; Aneboda, Gårdsjön, and Kindla, using a novel dynamic model, HD-MINTEQ (Löfgren et al., 2017), for the period 1880-2080. The HD-MINTEQ model is based on state-of-the-art descriptions of aluminum (Al) and base cation (BC) chemistry, and the version used in this paper incorporates the BC release kinetics from the PROFILE weathering model (Sverdrup and Warfvinge, 1993). First, we use historical data to model the soil chemistry dynamics from 1880 through 2080 assuming that there are no harvesting events in

the future. Modeled results are compared to measured soil water data for the period 1993 to 2010. Next we modeled the effects of two different harvesting intensities at 2020: CH, which removes stems only, and WTH, which in addition to stems also removes tops and branches. The objective of this work is to compare the impacts of CH, WTH, and no-harvest (NH) scenarios, as well as to compare the effects of harvesting with those observed for historic atmospheric deposition acidification. The parameters in focus are soil solution pH, soil solution BC concentration, $Ca^{2+}$ sorption, and weathering rates.

## 2. Methods

### 2.1 HD-MINTEQ

Simulations were run using the Husby Dynamic MINTEQ model (HD-MINTEQ), which connects the equilibrium calculations of Visual MINTEQ version 3.1 (Gustafsson, 2014), the Simple Mass Balance model (Sverdrup and De Vries, 1994), and the PROFILE model for soil chemical weathering (Sverdrup and Warfvinge, 1993). The details of HD-MINTEQ have previously

been described by Löfgren et al. (2017). In brief, it relies on the Stockholm Humic Model (SHM) for organic complexation (Gustafsson, 2001; Gustafsson and Kleja, 2005). The model assumes that the equilibria for ferrihydrite and $Al(OH)_3$ provide the upper limit for the solubility of $Fe^{3+}$ and $Al^{3+}$ in mineral soil horizons. Further, it uses an extended Freundlich model to simulate $SO_4$ adsorption (Gustafsson et al., 2015). HD-MINTEQ does not simulate N chemistry; instead dissolved $NH_4^+$ and $NO_3^-$ in the different horizons are given as input data. The soil pools of Al and organic C were assumed to be constant over the

simulated time period. To deal with water transport, HD-MINTEQ uses a 1-D advective-dispersive equation, although the actual dispersion is often governed by the thickness of the modelled soil layers. Vertical flow is assumed, which should be a reasonable approximation for the studied soils, as they were located in recharge areas where the soil surface was nearly flat. The plant uptake is distributed over the two or three uppermost layers, as in the SAFE model of Warfvinge et al. (1993).

### 2.2 Model setup

In the current application of the HD-MINTEQ model, the soils were compartmentalized into four different discrete horizons: an organic horizon (O), an eluvial layer (E), and two illuvial subsoil horizons (B1 and B2). Simulations were run over a 200-





year period from 1880 to 2080, with a 1-week time step. Three different forest stands were simulated; Aneboda (N 57° 05′ E 12° 32′), Gårdsjön (N° 58 40′ E° 12 30′), and Kindla (N 59° 05′ E 12° 01′) (Fig. S1), under three different harvest scenarios:

(1)   conventional harvest (CH, stems-only removal),

(2)   whole-tree harvest (WTH, 100% removal of above-ground logging residues), and

(3)   no-harvest control (NH).

At all three sites, the harvest events occurred in the year 2020.

The input data for each site and soil horizon are presented in Table 1 and were based on climate and soil profile (Podzol) data collected in earlier work. For the most part, these are given in Löfgren et al. (2011). The bulk densities were estimated as a function of organic C and soil depth using the empirical relationships of Nilsson and Lundin (2006). The volumetric water content was set to a common value of $0.3\ m^3\ m^{-3}$. Sensitivity analyses revealed that the model outcome was not greatly affected by the value of this parameter (data not shown). The extent of sulfate adsorption in the B1 horizon of Kindla was assumed to

be strongly sorbing ("strong" in Table 1), similar to that of another Podzol in the same area of Sweden (Gustafsson et al., 2015). The other two soils from south-western Sweden were assumed to have less (Table 1) sulfate adsorption in agreement with other soils from this area (Karltun, 1995); the relevant Freundlich parameters were taken from the Tärnsjö soil of Gustafsson et al. (2015). The mineral soil horizons were assumed to be in equilibrium with ferrihydrite, whereas Fe(III) was assumed to be negligible in the O horizons. Lysimeter data for dissolved organic C (DOC) were available for the time period

between 1993 and 2014. This means, however, that DOC was unknown for 179 of the 200 years in the chosen simulation period. Because the DOC is governed by a complex interplay of factors (e.g. Löfgren et al. 2010), it is difficult to predict DOC in a reliable way. Therefore for simplicity, DOC was assumed to be constant throughout the simulation period, and the values were calculated from the mean DOC concentrations between 1993 and 2014.

### 2.3 Deposition

Historical wet deposition data for the three sites (Fig. 1) were taken from Löfgren et al. (2011) and from Zetterberg et al. (2014). To calculate the contribution from dry deposition, results from measurements of a surrogate surface were used (Ferm and Hultberg, 1995; 1999). Reductions of the total deposition as a result of harvesting were considered using functions of Zetterberg et al. (2014). CH and WTH scenarios used the same deposition profiles, represented by the dotted-lines in Fig. 1. In the NH scenario, deposition values were maintained at 2019 levels from 2020 through 2080. The relatively high levels of

Na and Cl deposition at Gårdsjön were due to its proximity to the sea. The dips in deposition at Gårdsjön in the early 1900s and Kindla around 1890 were due to historical harvesting events. The dips that occur in 2020 were due to the simulated harvest scenarios. Following the rise and fall of $SO_4$ and $NO_3$ throughout the 20th century (grey shaded and labeled *Historical*




*acidification*) at all three sites, the influence of industrial emissions and subsequent regulations can be clearly seen. Currently, the $SO_4$ and $NO_3$ depositions are similar to the levels observed in the 1930s.

### 2.4 Plant uptake

BC uptake trends at Kindla and Aneboda (Fig. 2) were calculated as described previously (Zetterberg et al., 2014). Briefly,
biomass at any given time point was used to allocate the cation amount over time according to classical growth curves and information regarding any natural event (e.g., storms and fire) or silvicultural measures (e.g., clear-cutting and thinning) that have taken place during the rotation period. The final uptake curves were created by multiplying the biomass increments by the nutrient concentrations for various tree parts. BC uptake rates are based on current biomass (hindcast scenario) and future biomass predictions by the Swedish forest growth model ProdMod, version 2.2 (Ekö, 1985). BC uptake values were finally
corrected for litterfall (which returns BC to the soil) and remineralization estimates. The net BC uptake at Gårdsjön was estimated using the ForSAFE model, by dynamically simulating photosynthesis, growth and gross uptake in response to environmental drivers, and subtracting litterfall that was physiologically simulated in response to light saturation, respiration and water availability (Belyazid and Moldan, 2009).

The highest rates of net uptake occur in young growing forests, and as the forest stands age, less BC are taken up. The stack of graphs in the left column of Fig. 2 represents the NH scenario, which means that for Aneboda and Kindla, uptake trends after 2019 were modeled using an exponential decay and essentially approach zero as time goes on. The stack of graphs in the central and right columns (Fig. 2) represent the uptake trends that were used from 2020 and onwards under the CH and WTH scenarios, respectively. The negative values in uptake that occur after harvesting events are due to net influx of BC originating
from decomposition of leftover debris. The CH scenarios produce more negative values than WTH scenarios because there is more debris left on the soil. It should also be noted that the assumed 100% removal of harvest residues in the WTH scenario is an overestimation - in practice, ~70% is removed (Nilsson et al., 2015).

### 2.5 Mineralogy and weathering

For Aneboda and Kindla, the mineralogy used in the PROFILE submodel was calculated from data on total elemental analysis
using the A2M model (Posch and Kurz, 2007). For Gårdsjön, the mineralogy was taken directly from Martinson et al. (2003). The specific surface area used by PROFILE was estimated from the particle-size distribution using the relationship of Sverdrup (1996). The mineralogy data are presented in Table 1. To calculate weathering rates, the relationships of Sverdrup and Warfvinge (1993) were used. The only modification in the current work was the way in which the organic anion concentrations $[R^-]$ (mol $L^{-1}$) were calculated, as HD-MINTEQ uses the SHM whereas the original PROFILE model used the Oliver equation
(Oliver et al., 1983). Thus, the main equation to calculate the weathering rate is:



$$r = k_{H^+} \cdot \frac{\{H^+\}^{n_H}}{f_H} + \frac{k_{H_2O}}{f_{H_2O}} + k_{CO_2} \cdot P_{CO_2}^{n_{CO_2}} + k_R \cdot \frac{[R^-]^{n_R}}{f_R} \tag{1}$$

where $r$ = weathering rate (keq m$^{-2}$ s$^{-1}$) for an individual mineral, $k_{H^+}$, $k_{H_2O}$ and $k_R$ = rate coefficients for the reaction with

H$^+$, H$_2$O and DOC, respectively (m s$^{-1}$), $n_H$, $n_{CO_2}$, and $n_R$ = reaction orders of individual reactions, $P_{CO_2}$ = partial CO$_2$

pressure (atm), and $f_H$, $f_{H_2O}$, and $f_R$ = retardation factors ("brakes"). The latter are defined as:

$$f_H = \left(1 + \frac{[Al^{3+}]}{k_{Al}}\right)^{x_{Al}} \cdot \left(1 + \frac{2[Ca^{2+}] + 2[Mg^{2+}] + [K^+] + [Na^+]}{k_{BC}}\right)^{x_{BC}} \tag{2}$$

$$f_{H_2O} = \left(1 + \frac{[Al^{3+}]}{k_{Al}}\right)^{z_{Al}} \cdot \left(1 + \frac{2[Ca^{2+}] + 2[Mg^{2+}] + [K^+] + [Na^+]}{k_{BC}}\right)^{z_{BC}} \tag{3}$$

$$f_R = 1 + \left(\frac{[R^-]}{k_R}\right)^{0.5} \tag{4}$$

where $k_{Al}$, $k_{BC}$, and $k_R$ are saturation coefficients for dissolution reductions, whereas $x_{Al}$, $x_{BC}$, $z_{Al}$, and $z_{BC}$ are reaction orders. Values of all coefficients and reaction orders were taken from Sverdrup and Warfvinge (1993).

**2.6 Calibration**

To initialize the model, the following data for the first time step (in 1880) were provided: dissolved concentrations of major cations, anions and DOC, and for the solid phase geochemically active Al (Table 1). Based on this input, HD-MINTEQ then calculated the start pH and the corresponding sorbed concentrations of major ions as well as dissolved Al.

The model was calibrated in an iterative process in which geochemically active Al and the plant uptake percentages of the different layers were adjusted. Further it was assumed that the soil water chemistry was in steady-state with respect to the environmental conditions in 1880, i.e. with respect to the assumed values for atmospheric deposition, plant uptake, weathering, etc. Before calibration, an initial guess was made of geochemically active Al and of the base cation uptake percentages. During each iteration, the first step was to run the model for 1,000 years with the 1880 parameters to obtain initial (start-state) values

of dissolved ions. Final values produced after a 1,000-year simulation were then used to initiate the model, which was re-run





with historical disturbances and changes in atmospheric deposition. The modelled values of dissolved inorganic Al and other BC concentrations from 1993 to 2014 were then compared to soil solution data from the same period taken from various depths and binned into either horizon O, E, or B before being plotted (Löfgren et al., 2011). Further, it was checked that the assumed BC uptake percentages did not result in uneven depletion rates in the different layers; if so, the percentages were adjusted.

Based on the above, refined guesses of geochemically active Al and of the plant uptake distribution could be made for the next iteration.

## 3. Results and discussion

### 3.1 Historic acidification

Historic acidification of forest soils from atmospheric deposition of S took place from 1930 through 2000, and can be seen in

the modelled soil solution pH profiles at Aneboda (Fig. 3a), Gårdsjön (Fig. 4a), and Kindla (Fig. 5a). At Aneboda, the lowest soil solution pH for each horizon occurred around 1990, about the same time that atmospheric S deposition began to decline (Fig. 1a). The most dramatic decrease in pH at Aneboda occurred within the B1 horizon, dropping from pH 5.29 in 1940 to pH 4.81 in the late 80s, a total pH decline of 0.48 units in 47 years. The other horizons (O, E, and B2) at Aneboda saw milder changes in pH (<0.20 units) over the same time period. The effects of historical acid rain can also be seen in the BC profiles

of $Ca^{2+}$ and $Mg^{2+}$, most significantly in the B1 and B2 horizons of Aneboda (Fig. 3). An increase in dissolved BC occurred in order to balance the charge created by increasing levels of anions ($SO_4^{2-}$ and $NO_3^{-}$) being deposited from the atmosphere, i.e., these BC were exchangeable and responsive to anion concentrations. Between 1940 and 1980, dissolved $Ca^{2+}$ concentrations increased by 275% and 94% in the B1 and B2 horizon, respectively. The anomalous blip in pH (and BC) at Aneboda occurring between 2008 and 2018 is the result of a large storm that downed many trees in 2005, followed by a bark beetle infestation,

resulting in acidification and increased dissolved $Ca^{2+}$ (Löfgren et al., 2014). This may also explain the underestimated $SO_4^{2-}$ concentrations by the model (Fig. 3i), as the latter does not take into account the decomposition and oxidation of organically bound sulfur.

Of the three sites studied, the strongest effect of historic acid rain was seen at Gårdsjön (Fig. 4a), which experienced an average

decline of 0.39 pH units across all four horizons. This is likely due to the higher S (as sulfuric acid) deposition Gårdsjön (Fig. 1b). The horizon subject to the most drastic acidification was the E horizon, which experienced a sharp decline from pH 4.81 in 1940 to pH 4.02 by 1985 (a drop of 0.70 pH units in 45 years). Similar to Aneboda, Gårdsjön experienced its most acidic soil conditions in the mid 1980s. However, the B1 horizon did see a slight delay, experiencing its most acidic conditions in the early 1990s. This trend in delayed acidification across horizons was even more pronounced at Kindla (Fig. 5a), where the O

and E horizons reached their most acidic conditions by 1980, the B1 horizon continued to acidify until 1994, and the B2 horizon did not reach its most acidic conditions until 2013 (nearly 25 years after the S and N deposition began to decline, Fig.



1c). The delayed response was mainly attributed to $SO_4$ adsorption/desorption, which is known to delay response times to decades for strongly $SO_4$-adsorbing soil systems (Cosby et al., 1986).

The dissolved aluminum profiles at all three sites show that most of the mobilization of dissolved Al occurred in the E horizon.
This was likely due to the higher pH of the B horizons (pH > 5), where the precipitation of aluminum hydroxide and/or allophane removed soluble Al (Gustafsson et al., 1995; Karltun et al., 2000). This inverse relationship with pH is demonstrated clearly for Kindla's B1 horizon, where soluble Al concentration peaked as pH decreased below 5 (Fig. 5).

### 3.2 Harvesting-induced acidification

One trend across all horizons at all sites was that directly after a harvesting event (CH or WTH) in 2020, the pH sharply increased for 5-10 years before slowly declining over the remainder of the simulation (displayed in Figs. 3b, 4b, and 5b). The increase in pH immediately following a harvest event is primarily caused by mineralization of BC-containing harvest residues (Fig. 2), but also marginally by a decreased dry S deposition. These processes increase the sorption of BC on the exchange sites (Figs. 3l, 4l and 5l). The impact of harvesting on pH and sorbed Ca lasted for three to four decades at all three sites,
eventually converging towards the NH scenario, before falling below it, which was demonstrated previously in long-term experiments (Zetterberg et al., 2016). However, once new stands have been established and trees begin to grow, uptake drives a net loss of BC from the soil, leading to acidification. Across all three sites, the B1 horizons became the most acidified by 2080 compared to the NH scenario. The O horizon experienced the least change after harvesting events, and at Aneboda and Kindla, the pH was slightly *less* acidic compared to the NH scenario by 2080. However, had the simulation been run for a
longer time, it appears that the pH would eventually reach NH levels, or drop below them. In reality a second harvest would likely have occurred after some 80-100 years, yielding a new period with a brief pH increase.

Comparing the two harvesting scenarios demonstrates that over the 60 year time-frame studied, WTH acidified the soil more than CH, but not by much even though 100% of the harvest residues are assumed withdrawn at WTH. At Aneboda, the pH
across all horizons dropped by an average 0.13 (WTH) and 0.12 units (CH) compared to the NH scenario by the year 2080. The difference between WTH and CH was more pronounced at Kindla, which saw a mean pH decrease across all horizons of 0.10 (WTH) and 0.04 (CH) units compared to the NH scenario by 2080. The most sensitive horizon to harvesting at Aneboda was the B1 horizon, which dropped in pH by 0.42 (WTH) and 0.36 (CH) units compared to the NH scenario by the year 2080. Aneboda's B1 horizon also experienced a significant loss of soluble $Ca^{2+}$ after harvesting, decreasing by 65% (WTH) and 56%
(CH) by 2080 compared to the NH control. However, on a percentage basis, Aneboda's E horizon experienced the most precipitous loss of soluble $Ca^{2+}$ after harvesting, decreasing by 87% (WTH) and 86% (CH) by 2080 compared to the NH control. Similar trends were seen in sorbed $Ca^{2+}$ concentrations at Aneboda by 2080. The E horizon experienced a 92% (WTH) and 91% (CH) decrease compared to the NH control, while the same figures for the B1 horizon were 83% and 75%, respectively. Neither harvesting scenario appeared to have a tangible impact on the $Ca^{2+}$ concentrations (soluble or sorbed) in



the B2 horizon at Aneboda. Contrary to the trends in $Ca^{2+}$ concentrations in horizons E, B1, and B2, after harvesting, the O horizon experienced an increase in soluble and sorbed $Ca^{2+}$ by 2080 compared to the NH control.

Compared with the NH scenario, Kindla also experienced a phase with an increased pH followed by acidification in the end of the simulation period, but to a lesser degree than Aneboda. At Kindla, the mean soil pH at 2080 decreased by an average of 0.04 and 0.10 units after CH and WTH, respectively. The B1 horizon was the most sensitive to acidification in both harvesting scenarios. Of the Kindla horizons, B1 also saw the greatest change in soluble and sorbed $Ca^{2+}$ after harvesting. Soluble $Ca^{2+}$ decreased by 80% (WTH) and 73% (CH), while sorbed $Ca^{2+}$ decreased 91% (WTH) and 84% (CH) by 2080 compared to the NH scenario. One interesting observation at Kindla was that the delayed acidification with increasing soil depth, which was related to $SO_4^{2-}$ adsorption/desorption processes and historic acidification, was not observed after harvesting. Our simulations indicate that biological acidification do not initiate such processes at least not within 60 years after harvest, corresponding to an almost full rotation period.

A pH increase after harvesting was also observed at Gårdsjön, with the O and E horizons experiencing the strongest response. After increasing for a couple of years after harvest, the pH started to decline, slowly approaching the NH scenario. By the end of the simulation (2080), there was almost no difference between CH, WTH, and the NH scenarios, and contrary to the trends at Aneboda and Kindla, it did not appear that the soils were trending towards further acidification post-2080. This trend of disturbance immediately following a harvest before slowly approaching the NH trends can also be seen in BC concentrations. Total dissolved Al, Ca, and Mg sharply decreased immediately after CH and WTH for several years before meandering towards the NH trend line. However, by 2080, the concentration of soluble BC after WTH was significantly lower than after CH, and both harvesting events resulted in less soluble BC compared to the NH scenario. Exchangeable $Ca^{2+}$ concentrations following harvesting events at Gårdsjön appeared to oscillate around those of the NH scenario: increasing immediately after harvest (similar to Aneboda and Kindla), before slowly decreasing (also similar to Aneboda and Kindla), but then again increasing to approach the NH levels of sorbed $Ca^{2+}$. There are several possible explanations for the $Ca^{2+}$ profiles at Gårdsjön. First, Gårdsjön has a relatively high organic carbon content (Table 1), which governs the cation exchange capacity, resulting in higher exchangeable $Ca^{2+}$ to buffer the soil pore water. Also, atmospheric deposition at Gårdsjön was significantly higher than at the other locations (Fig. 1) resulting in a higher ionic strength pore water, which would result in quicker transport of soluble ions between the horizons.

The results indicate that the acidification due to a harvesting event at 2020 was less impactful, over the time range studied, than that of historic atmospheric acidification. Even though the pH in the WTH and CH scenario decreased with time as compared to the NH scenario, the pH did not drop to the levels observed around the peak of historic acidification (1980-1990). As was discussed by Löfgren et al., (2017), this reflects the different acidification mechanisms involved. Most importantly,





the concentration of mobile anions was much lower in the harvesting scenarios compared to the levels around the 1980s and 1990s, and this limits the potential pH decrease.

Moreover, the drop in pH after 2020 at Aneboda and Kindla was limited by the amount of sorbed $Ca^{2+}$, restricting $Ca^{2+}$ available

for vegetation uptake. In the harvesting scenarios the sorbed $Ca^{2+}$ pool was not replenished in spite of the increased pH after historic acidification, which is similar to what was observed in the MAGIC simulations of Zetterberg et al. (2014). This in turn led to extremely low levels of dissolved $Ca^{2+}$ and to occasional model errors due to negative concentrations in preliminary model runs. To avoid this from happening, the net $Ca^{2+}$ uptake was decreased as dissolved $Ca^{2+}$ became low, according to the following relationships taken from the "old" SAFE model (Alveteg, unpublished):

$$Ca_{upt,real} = f_{upt} \cdot Ca_{upt,init} \qquad (5)$$

where $Ca_{upt,\,real}$ is the adjusted net Ca uptake, $Ca_{upt,\,init}$ is the assumed Ca uptake according to Table 1, and $f_{upt}$ is defined by:

$$f_{upt} = \frac{[Ca^{2+}]^n}{[Ca^{2+}]^n + c^n} \qquad (6)$$

where $n = 4$ and $c = 1.5 \cdot 10^{-6}$. These equations prevented dissolved $Ca^{2+}$ concentrations from falling below 5 µmol L$^{-1}$, but it also limited the acidification effect due to Ca uptake. To what extent trees will adjust their BC uptake in response to lower availability in the soil is still debated (Zetterberg et al., 2014). In plot experiments in which the effects of harvesting were

studied, the depletion of sorbed $Ca^{2+}$ was less severe than that predicted by mass-balance calculations and models (Zetterberg et al., 2014). Possible reasons include both a lower $Ca^{2+}$ uptake to trees and mineralization of strongly bound $Ca^{2+}$ in litter, which is otherwise not geochemically active. However, because the HD-MINTEQ simulations here produced stronger effects on soil Ca compared to those observed in the field (Zetterberg et al., 2016), it seems probable that the acidification effect as predicted by HD-MINTEQ may be regarded as a "worst-case scenario", and that the real acidification, at least over the first

rotation period, may be even smaller. However, at this point it needs to be added that these conclusions may not be relevant in cases when nitrification following harvest is substantial, in which case the acidification effect could be considerably larger; this possibility was not considered in our simulations.

### 3.3 Weathering and release of base cations

The weathering rates as calculated by the PROFILE submodel in HD-MINTEQ differed across locations and layers in response to historic acidification and the presence of weathering brakes such as $Ca^{2+}$ and $Al^{3+}$ (Eqs. 2 and 3). Across all sites and layers,





the major BC released by weathering were Ca and Na (Fig. 6). During the historic acidification period, the annual weathering rates at several sites and horizons (Aneboda: E; Gårdsjön: E, B1, and B2; Kindla: B1 and B2) actually decreased (Fig. S2). This is due to the brakes in the weathering function (i.e. from $Al^{3+}$ and BC, Eqs. 2 and 3) and to the fact that total dissolved Al and BC were much higher during this period (Figs. 3, 4 and 5). However, the exact patterns varied from site to site, and from

layer to layer. The low weathering rates in the E horizon at Gårdsjön were likely due to its mineralogy, i.e., minerals other than quartz, K-feldspar, and plagioclase were essentially absent from the E horizon, but they were present further down in the profile. Another factor leading to low weathering rates in the Gårdsjön E horizon was the relatively thin layer thickness.

Contrary to the decrease in weathering rates during the historic acidification of the 1970s, the weathering rates after the
harvesting scenarios at 2020 generally increased compared to NH by 2080 (Fig. 7). However, the dynamics were quite different across each layer and site. The B1 horizon at Aneboda saw the strongest increase in weathering rates after harvesting, increasing by 9% (CH) and 11% (WTH) by 2080 compared to the NH scenario. The other horizons at Aneboda (E and B2) saw relatively small increases in weathering rates (<2% by 2080 compared to a NH scenario). It is worth noting that the difference between harvesting events and NH at the Aneboda B1 and B2 horizons would likely continue to diverge beyond
2080. At Aneboda, by the year 2080, the sum of weathered BC for the CH and WTH scenarios were 32 (CH) and 46 (WTH) meq $m^{-2}$ higher, respectively, than the BC weathered in the NH scenario (Fig. S3).

The trends in weathering rates at Gårdsjön after harvesting mimicked the trends seen in pH, i.e., they rapidly increased for a couple of years, before declining to approach similar levels to the NH scenario. For all three layers at Gårdsjön, the weathering
rates appeared to reach steady-state conditions by the year 2055, whereas the weathering rates at Kindla and Aneboda deviated from the NH trendlines well into 2080. On a mass basis, CH and WTH at Gårdsjön resulted in amounts of weathered BC that were 74 and 49 meq $m^{-2}$ higher than in the NH scenario by the year 2080 (Fig. S3).

At Kindla, no horizon saw an increase in the weathering rate greater than 3.4% by 2080, compared to the NH scenario. In fact,
when weathering of the three mineral soil layers was summed, weathering rates slightly decreased by 0.9% (CH) and 2.0% (WTH) by 2080 compared to the NH scenario, which was equivalent to a decreased amount of weathered BC of 64 and 26 meq $m^{-2}$, respectively, over the simulated time period (Fig. S3). This result was influenced mainly by the large decrease in weathering rates in the E horizon in response to the pH increase after harvest (Figure 7f).

Weathering is dictated by both $H^+$ (the lower the pH, the more weathering) and by the weathering brakes, most importantly dissolved $Al^{3+}$ (the more $Al^{3+}$, the less weathering). Some layers followed the generally expected trend of increased weathering rates with lower pH (B1 and B2 of Aneboda, and E of Kindla). These same layers were also low in dissolved $Al^{3+}$, as shown by log $\{Al^{3+}\}$ (Fig. S4), so the $Al^{3+}$ weathering brake (the first terms of Eq. 2 and Eq. 3) was not strong, and as a result the $H^+$ concentration dictated mineral weathering. In all other layers (E of Aneboda, all horizons in Gårdsjön, and both B horizons of



Kindla) the weathering rates *decreased* with lower pH, which was due to a relatively high concentration of dissolved $Al^{3+}$ that caused the $Al^{3+}$ weathering brake to be important. There is a certain threshold range where the weathering brake due to $Al^{3+}$ concentration started to influence weathering rates greatly, rendering the $H^+$ activity less important. The "weathering rate vs. Al" profiles in Figure S4c show the transition from pH-controlled weathering to Al-controlled weathering —weathering rates increase with increasing log $\{Al^{3+}\}$ until a "break point" $Al^{3+}$ activity is reached, where weathering rates begin to level off and decline. These results are consistent with those of Erlandsson et al. (2016), who showed that PROFILE predicts a negligible importance of the Al weathering brake at low values of log $\{Al^{3+}\}$, but at log $\{Al^{3+}\}$ > -7 the brake becomes increasingly influential particularly for the feldspars, which were important mineral constituents at all three sites in the current study (Table 1). There are other weathering brakes as well, such as $Ca^{2+}$, that limit the effect of pH-induced weathering (Eq. 2), but these were less important (data not shown). Although not obvious at first, the trends in weathering rates within an individual layer were consistent, regardless of the source of acidification (be it acid rain or harvesting).

## 4. Conclusions

Forest soils are no longer threatened by acid deposition. There is, however, the potential for forest harvest to increase soil acidification. Based on simulations of three forested sites in Sweden, acidification due to harvesting events (even a whole-tree harvest including 100% of the harvest residues) had much less impact on soil pH compared to historical acid rain, provided that harvesting did not cause substantial nitrification (which was not considered in the model). The strongest effect of historic acid rain was seen at Gårdsjön, the site that experienced the most sulfuric acid deposition. Furthermore, our model results suggest that during the historic acidification era, there was a decrease in weathering rates at some locations mostly due to high dissolved Al concentrations in soil solution.

Although the long-term pH effect of harvest was predicted to be small in relation to historical acidification, future harvesting did significantly change the soil chemistry. Directly after a harvest, soil solution pH increased and remained above the NH baseline for several decades before eventually dropping below the baseline. Decomposition of harvest residues, which liberate base cations, and a decreased dry sulfur deposition, were likely responsible for the alkalization immediately following a harvesting event. Over a period of 60 years after conventional harvest, BC weathering increased by 32 and 74 meq m$^{-2}$ as compared to the no-harvest scenario for the Aneboda and Gårdsjön sites, respectively. Over the same time period, Kindla saw a decreased BC weathering. These diverging results were explained by differences in the relative significance of pH-driven weathering and by the inhibitory effect of dissolved $Al^{3+}$. In soil layers with low dissolved $Al^{3+}$, the weathering rates increased with decreased pH, whereas the opposite was usually observed for layers high in dissolved $Al^{3+}$.

*Data and code availability.* Data on soil and solution chemistry are available from the Integrated Monitoring website at http://info1.ma.slu.se/im/IMeng.html. The HD-MINTEQ code can be made available upon request to gustafjp@kth.se.



*Competing interests.* The authors declare that they have no conflict of interest.

*Special issue statement.* This article is part of the special issue "Quantifying weathering rates for sustainable forestry"
(BG/SOIL inter-journal SI). It is not associated with a conference.

**The Supplement related to this article is available at**

*Acknowledgements.* This study was funded by the Swedish Research Council Formas (reg. no. 2011-1691) within the strong
research environment "Quantifying weathering rates for sustainable forestry (QWARTS)" and by the Swedish Energy Agency
(project no. 31708-3). Carin Sjöstedt is acknowledged for help with the uptake scenarios.

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



**Table 1.** Parameter values and assumptions used for the HD-MINTEQ simulations of all three sites.

| Site | Aneboda | | | | Gårdsjön | | | | Kindla | | | |
|---|---|---|---|---|---|---|---|---|---|---|---|---|
| Horizon | O | E | Bs1 | Bs2 | O | E | Bs1 | Bs2 | O | E | Bs1 | Bs2 |
| Thickness (m) | 0.1 | 0.1 | 0.15 | 0.15 | 0.05 | 0.05 | 0.2 | 0.15 | 0.1 | 0.1 | 0.15 | 0.15 |
| Bulk density (kg m$^{-3}$) | 118 | 837 | 980 | 1122 | 156 | 773 | 749 | 836 | 118 | 1004 | 980 | 1237 |
| Volumetric water content (m$^3$ m$^{-3}$) | 0.3 | 0.3 | 0.3 | 0.3 | 0.3 | 0.3 | 0.3 | 0.3 | 0.3 | 0.3 | 0.3 | 0.3 |
| Dispersivity (m) | 0.01 | 0.01 | 0.01 | 0.01 | 0.01 | 0.01 | 0.01 | 0.01 | 0.01 | 0.01 | 0.01 | 0.01 |
| Discharge (mm) | 500 | 400 | 350 | 350 | 550 | 500 | 450 | 450 | 600 | 500 | 450 | 450 |
| Winter soil temp (°C) | 3 | 4 | 6 | 8 | 3 | 4 | 6 | 8 | 3 | 4 | 5 | 6 |
| Summer soil temp (°C) | 12 | 11 | 9 | 8 | 12 | 11 | 9 | 8 | 10 | 8 | 7 | 6 |
| Dissolved NH$_4^+$ (μM) | 14 | 14[a] | 5.5[a] | 5.5[a] | 0.5 | 2.6 | 4.2 | 4.2 | 0.5 | 2.9 | 3.9 | 3.9 |
| Dissolved NO$_3^-$ (μM) | 0.5 | 0.5[a] | 0.3[a] | 0.3[a] | 0.5 | 0.4 | 0.4 | 0.4 | 0.5 | 0.3 | 0.5 | 0.5 |
| Organic C (%) | 40 | 4 | 2.5 | 1.5 | 40 | 5 | 4.5 | 4.5 | 40 | 2 | 2.5 | 0.8 |
| Sulfate adsorption | no | no | no | some | no | no | some | some | no | no | strong | some |
| Equilibrium with ferrihydrite | no | yes | yes | yes | no | yes | yes | yes | no | yes | yes | yes |
| Geochemically active Al (mmol kg$^1$) | 30 | 35 | 32 | 30 | 40 | 50 | 80 | 80 | 30 | 15 | 60 | 60 |
| Partial CO$_2$ pressure (atm) | 0.001 | 0.003 | 0.007 | 0.01 | 0.001 | 0.002 | 0.007 | 0.01 | 0.001 | 0.003 | 0.007 | 0.01 |
| DOC (mg L$^{-1}$) | 50 | 48.5 | 7.7 | 7.7 | 35 | 12.6 | 9.8 | 9.8 | 25 | 13.7 | 3.7 | 3.7 |
| Base cation uptake (% of total) | 20 | 50 | 30 | 0 | 50 | 30 | 20 | 0 | 20 | 50 | 30 | 0 |
| Start of growth period (week of the year) | 15 | 15 | 15 | 15 | 15 | 15 | 15 | 15 | 20 | 20 | 20 | 20 |
| Duration of growth period (weeks) | 30 | 30 | 30 | 30 | 30 | 30 | 30 | 30 | 20 | 20 | 20 | 20 |
| | | | | | | | | | | | | |
| Mineral composition | | | | | | | | | | | | |
| K-Feldspar (%) | - | 5.76 | 7.52 | 14.11 | - | 15.0 | 18.0 | 19.0 | - | 12.2 | 10.75 | 10.74 |
| Plagioclase (%) | - | - | - | - | - | 14.0 | 15.0 | 16.0 | - | - | - | - |
| Anorthite (%) | - | 3.63 | 2.24 | 4.20 | - | - | - | - | - | 1.63 | 0.64 | 1.10 |
| Albite (%) | - | 22.41 | 25.69 | 27.49 | - | - | - | - | - | 27.81 | 21.22 | 23.08 |
| Hornblende (%) | . | 1.11 | 1.01 | 1.62 | - | 0.1 | 1.5 | 1.5 | - | 1.76 | 0.33 | 0.95 |
| Epidote (%) | - | 4.56 | 4.07 | 4.13 | - | 0.1 | 0.75 | 1.0 | - | 2.14 | 0.61 | 1.30 |
| Garnet (%) | - | - | - | - | - | 0.1 | 0.1 | 0.1 | - | - | - | - |
| Biotite (%) | - | - | - | - | - | 0.5 | 0.5 | 0.5 | - | - | - | - |
| Apatite (%) | - | 0.09 | 0.24 | 0.24 | - | 0.1 | 0.2 | 0.3 | - | 0.21 | 0.09 | 0.15 |
| Fe-Chlorite (%) | | | | | - | 0.4 | 0.4 | 0.4 | - | - | - | - |
| Chlorite (%) | - | 0.50 | 0.40 | 0.88 | - | - | - | - | - | 0.81 | 0.19 | 0.33 |
| Illite1 (%) | - | 2.07 | 3.53 | 1.77 | - | - | - | - | - | 1.43 | 1.24 | 4.35 |
| Mg-Vermiculite (%) | - | - | - | - | - | 3.0 | 15.0 | 5.0 | - | - | - | - |
| Vermiculite 1 (%) | - | 1.21 | 2.16 | 2.78 | - | - | - | - | - | 1.64 | 0.48 | 0.84 |
| Vermiculite 2 (%) | - | 0.50 | 0.38 | 0.84 | - | - | - | - | - | 0.99 | 0.16 | 0.31 |
| Muscovite (%) | - | 6.92 | 10.67 | 3.65 | - | - | - | - | - | 1.56 | 1.36 | 8.06 |
| Rutile (%) | - | 0.56 | 0.49 | 0.49 | - | - | - | - | - | - | 0.21 | 0.28 |
| Hematite (%) | - | 2.33 | 0.73 | 0.73 | - | - | - | - | - | 0.43 | 0.08 | 2.11 |
| Chlorite1 (%) | - | 0.73 | 0.63 | 1.72 | - | - | - | - | - | 1.35 | 0.33 | 0.49 |
| Specific surface area (m$^2$ g$^{-1}$) | - | 1.04 | 0.95 | 0.95 | - | 1.2 | 1.1 | 1.98 | - | 1.29 | 1.18 | 1.18 |

[a]These values were adjusted for calibration to account for elevated N concentrations detected in lysimeter data from 2011 onwards. This was due to storm damage (Gudrun) in the region, which brought down many trees in 2005.





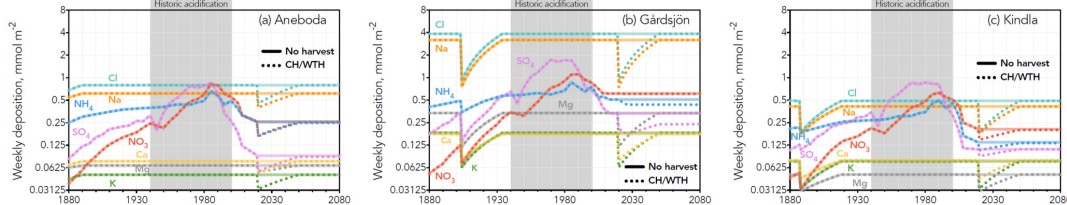

**Figure 1.** Weekly ion deposition values (mmol m$^{-2}$) used for WTH/CH and NH simulations at Aneboda (a), Gårdsjön (b), and Kindla (c).




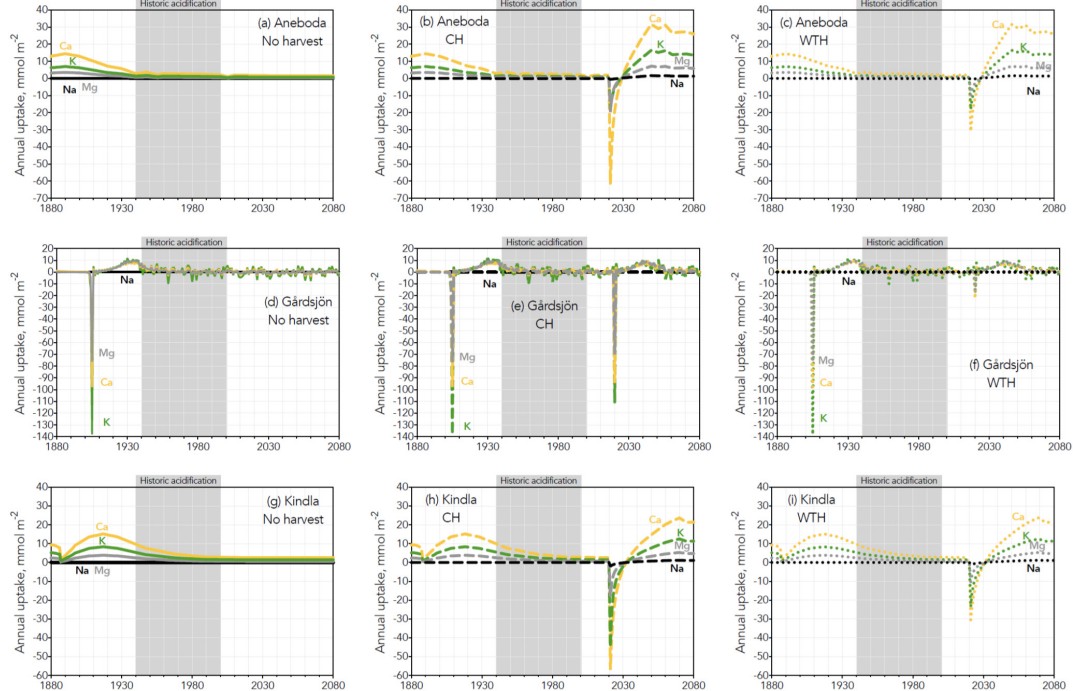

**Figure 2.** Annual net base cation uptake trends at Aneboda (a–c), Gårdsjön (d–f), and Kindla (g–i) under NH (a, d, and g),

5  CH (b, e, and h), and WTH (c, f, and i) scenarios. The harvest events (CH and WTH) occurred in 2020.



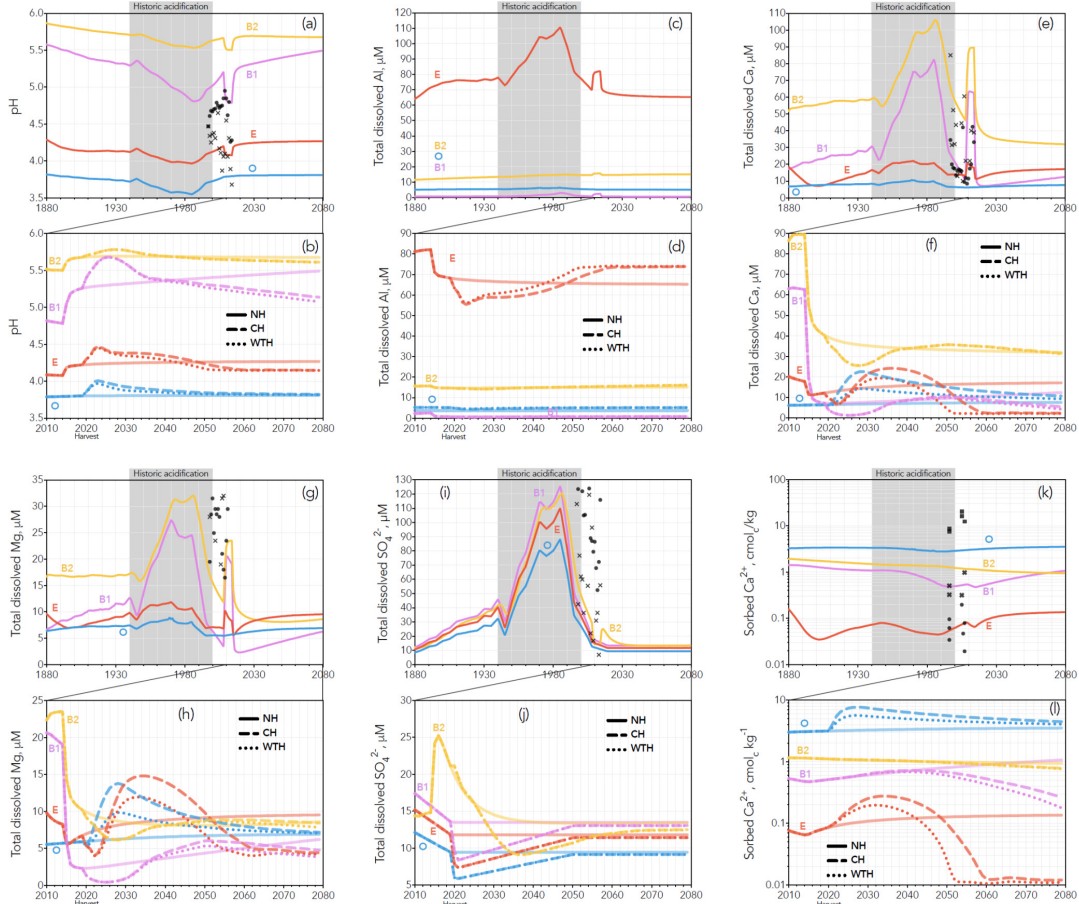

**Figure 3.** Aneboda. Simulated mean annual pH (a and b), total dissolved Al (c and d), Ca (e and f), Mg (g and h), and SO$_4^{2-}$ (i and j), and sorbed Ca$^{2+}$ (k and l). Chemical dynamics in response to historic acidification (a, c, e, g, i, and k) are presented above the response to harvesting events (b, d, f, h, j, and l). The solid, dashed, and dotted lines represent the annual averages in the NH, CH, and WTH scenarios, respectively. The grey box spanning 1940 to 2000 represents the period of acidification due to S deposition. The symbols "■", "•", and "▯" represent the annual averages of empirical measurements in the O, E and B horizons, respectively.




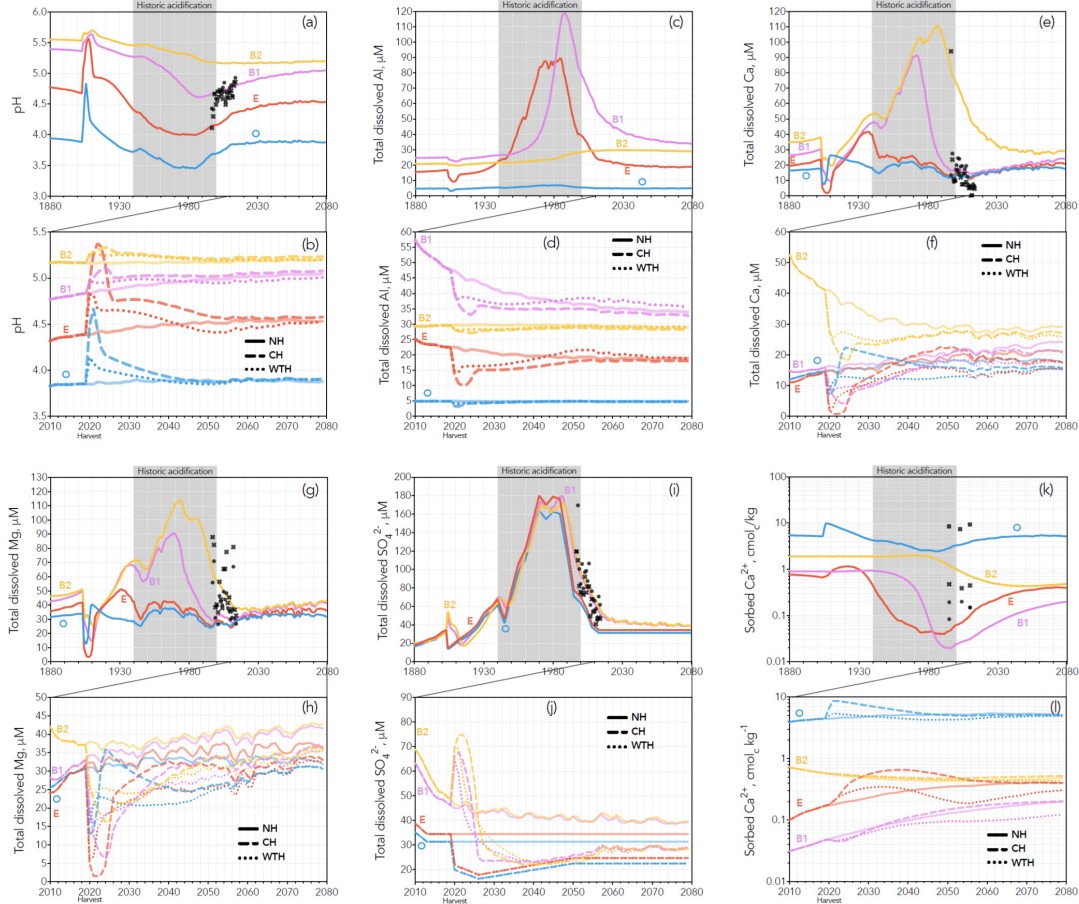

**Figure 4.** Gårdsjön. Simulated mean annual pH (a and b), total dissolved Al (c and d), Ca (e and f), Mg (g and h), and SO$_4$$^{2-}$ (i and j), and sorbed Ca$^{2+}$ (k and l). Chemical dynamics in response to historic acidification (a, c, e, g, i, and k) are presented above the response to harvesting events (b, d, f, h, j, and l). The solid, dashed, and dotted lines represent the annual averages in the NH, CH, and WTH scenarios, respectively. The grey box spanning 1940 to 2000 represents the period of acidification due to S deposition. The symbols "■", "•", and "▢" represent the annual averages of empirical measurements in the O, E and B horizons, respectively.



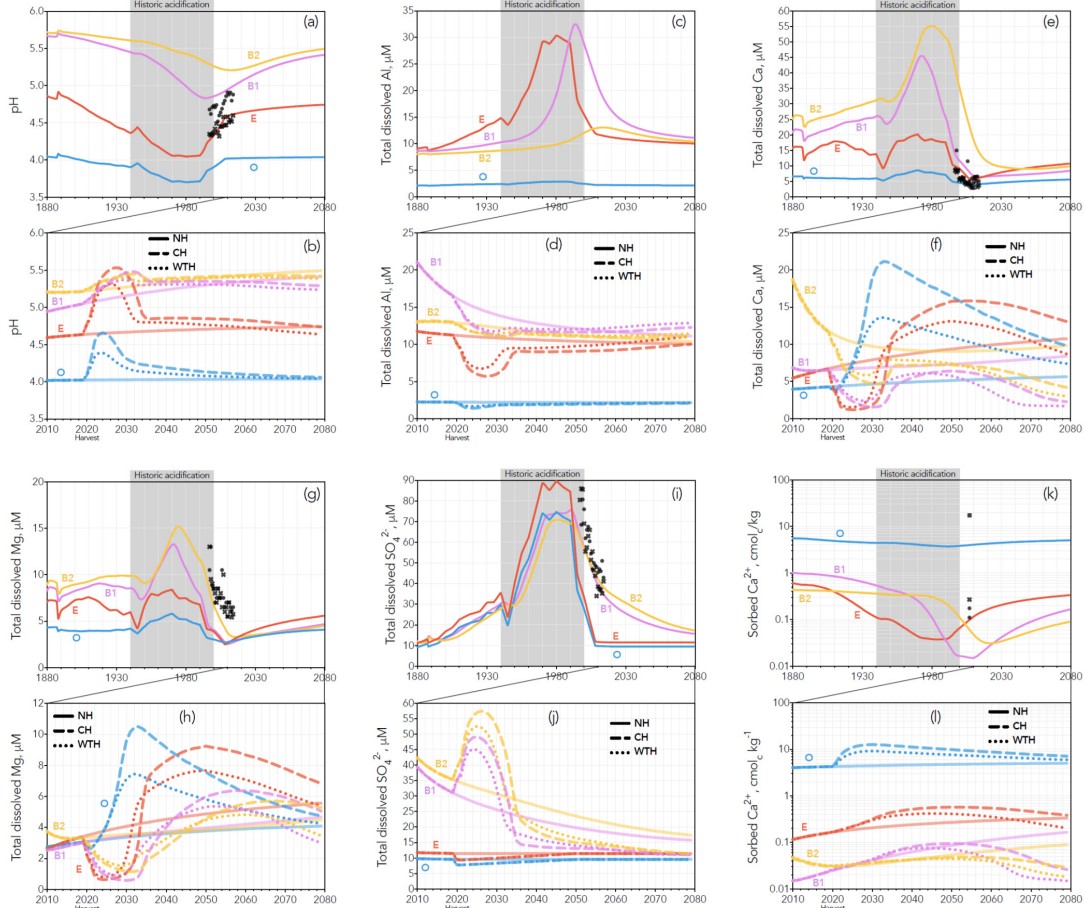

**Figure 5.** Kindla. Simulated mean annual pH (a and b), total dissolved Al (c and d), Ca (e and f), Mg (g and h), and SO₄²⁻ (i
and j), and sorbed Ca²⁺ (k and l). Chemical dynamics in response to historic acidification (a, c, e, g, i, and k) are presented
above the response to harvesting events (b, d, f, h, j, and l). The solid, dashed, and dotted lines represent the annual averages
5 in the NH, CH, and WTH scenarios, respectively. The grey box spanning 1940 to 2000 represents the period of acidification
due to S deposition. The symbols "■", "●", and "▢" represent the annual averages of empirical measurements in the O, E and
B horizons, respectively.

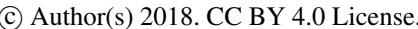



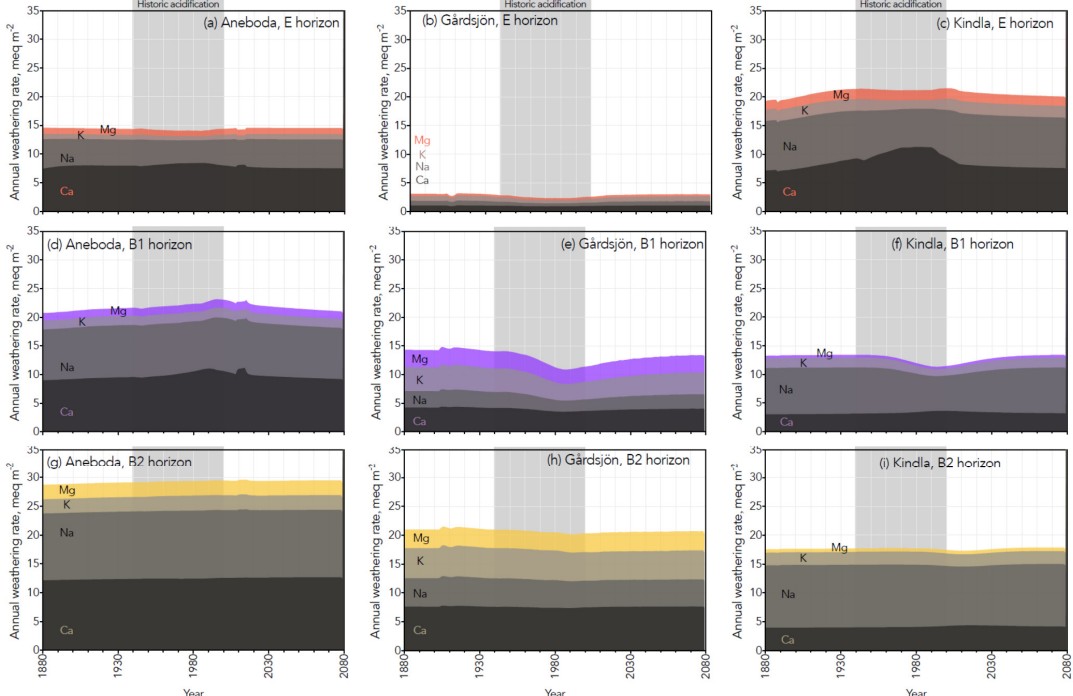

**Figure 6.** Stack plot of simulated base cation weathering rates in the NH scenario, with subdivisions of elemental components.





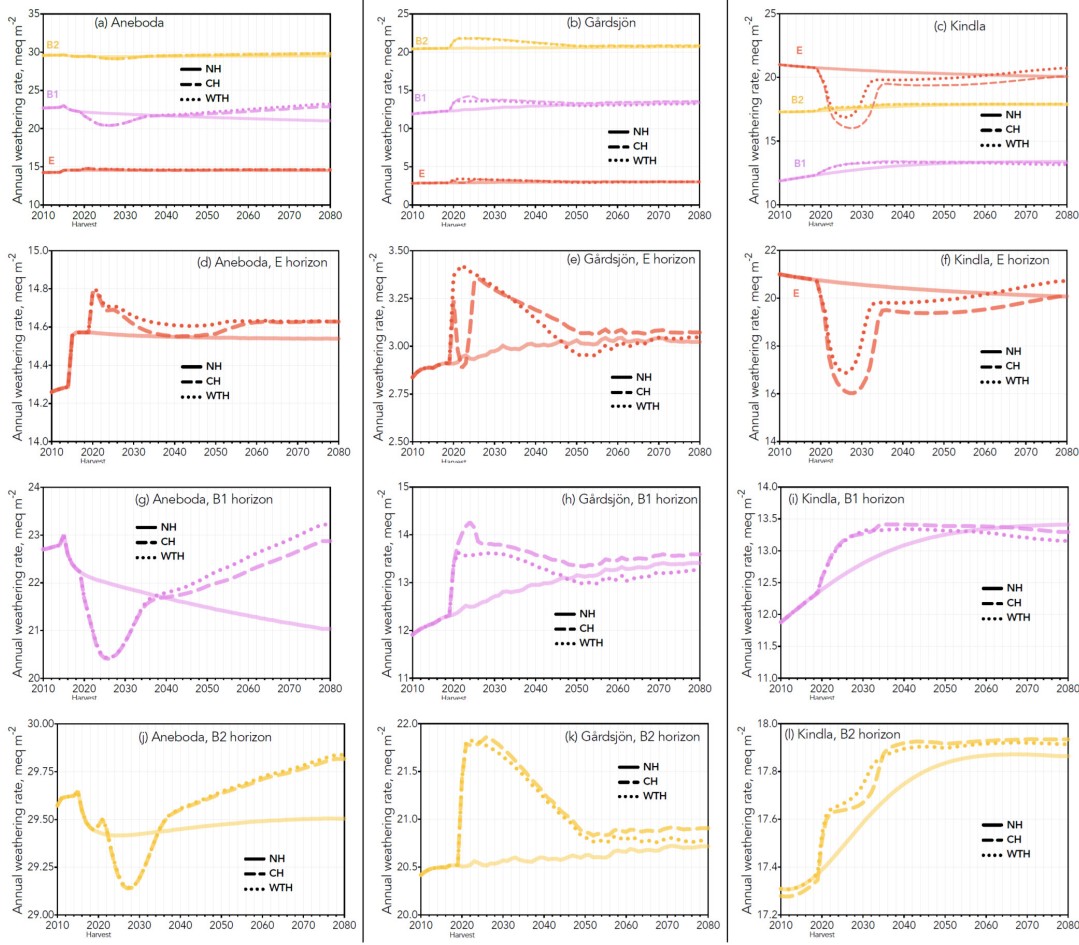

**Figure 7.** Simulated effects on base cation weathering rates in the conventional harvest (CH), whole-tree harvest (WTH) and NH scenarios. The time series starts in 2010 just before harvest and the graphs are scaled in order to demonstrate differences

5   between harvest intensities at the different sites and soil horizons.