# Peer review of "Assessing the impact of acid rain and forest harvest intensity with the HD-MINTEQ model – Soil chemistry of three Swedish conifer sites from 1880 to 2080"

_SOIL, 2018_

## Referee Comment (RC1) · Anonymous Referee #1 · 23 Aug 2018

General comments

The authors present a modelling study of the past effects of acid deposition and projected future effects of varying harvesting practices on the soil and soil solution chemistry of three forested sites in Sweden. This is clearly a topic of current interest as the impacts of past soil acidification recede and the potential future impacts of forest management practice become more important. The manuscript is well organised and written and is reasonably straightforward to understand. Having read the paper, I do however feel that significant amendment is needed, the reasons for this being largely

to do with the broader context of the work and the manner in which the results are presented and discussed:

1. it is not clear exactly what the purpose of the study is. Is it to demonstrate the utility of the HD-MINTEQ model for the projection of future soil chemistry under different forest management practices, to specifically make projections that can inform forest management practice, or a combination of both? The manner in which the paper is written suggests largely the latter, in which case I have reservations regarding the confidence that can be placed in the model as currently constructed as a comprehensive predictor of soil and soil solution chemistry. The lack of a specific submodel for N species transformations is in my opinion a potentially significant shortcoming.

2. It is not clear what specifically is new about this work in the context of modelling soil and soil solution chemistry. The model is state of the art for its description of ion-organic matter equilibrium interactions and the description of BC weathering, but it falls somewhat short of the state of the art in other areas.

3. Given these acknowledged shortcomings of the model, it would seem appropriate to me to place more emphasis than has been placed, on assessing the ability of the model to predict the trends and/or magnitudes in soil and soil solution chemistry. The results and discussion section effectively takes the model predictions as 'correct' in its assessment of future trends.

4. I feel that the paper would be strengthened by an assessment of the model capabilities alongside the discussion of the future projections, coupled with an assessment of where the model could be strengthened e.g. a better description of N transformations (or, of course, arguments as to why such 'improvements' would not greatly improve predictions!).

5. Finally (and returning somewhat towards my first bullet point) I cannot see any broader context - why is there a need to make these predictions in the first place? For example, can the results tell us anything about the effects of different future management practices on forest viability? I appreciate that this is somewhat beyond the scope of this paper as it relates to relationships between soil chemistry and tree growth/performance, but I do feel that some broader context is needed to emphasise the importance and usefulness of this work.

Specific comments

p3 line 23. How is the supply of N species concentrations realised over the whole simulation period? It is currently unclear how exactly N species are handled. p3 line 24. It is not clear what is meant by 'The soil pools of Al and organic C were assumed to be constant over the simulated time period. Does this mean that losses of Al and C in drainage are assumed to be completely replenished? Does the soil pool of Al mean the geochemically active pool, or is there a non-reactive component that supplies active Al by weathering? p5 line 10. Why was ForSAFE only used at Gårdsjön?

p7, line 4. It is unclear what is meant by uneven depletion rates or how the BC uptake rates were adjusted to compensate for this.

Section 2.6. My interpretation is that pH was not used in calibration? This seems unusual given the importance of pH changes as a consequence of soil management and acidification/recovery. Why was this not done?

Section 3 (Results and discussion). There appears to be no comparison of the model predictions of pH, BC and sulphate with the observations. This to me is a serious omission since the usefulness of the model in predicting the future effects of forest management depends critically on its ability to predict how the soil chemistry has responded to changing inputs and management in the past. Does the model predict the trends and/or magnitudes of the observations over time to a degree that provides sufficient confidence for it to be a useful tool for investigating the projecting future trends? If not, can potential reasons for discrepancies be suggested and means to further develop the model to address such reasons be proposed?

p10, line 4 onwards. As an explanation of a modification to the model in the light of initial runs, this discussion would be better placed close to the model calibration section.

Figures 3-5. It is somewhat difficult to follow the graphs but my impression is that pH in B1&B2 is modelled as being consistently higher than pH in E, but that the observations suggest that pH is either quite similar in both horizons or tends to be higher in E. This is an example where the model capabilities need to be assessed against the observations - is the model a useful tool for assessing what is being projected here?

Technical corrections

Figures 3-5. It is rather difficult to match observations and model results on those panels where observations are present. I suggest redrawing to make the connection more obvious - a simple way would be to colour the observation points according to horizon in the same way as the model lines (though the B1/B2 lines might need to be recoloured also).

Figures 3-5. The symbols noted in the captions do not correspond with those on the graphs - specifically the graphs have crosses, which are not listed in the captions. This needs correction. I've assumed that crosses are supposed to be open squares.

---

## Referee Comment (RC2) · Anonymous Referee #2 · 25 Sep 2018

In their manuscript, the authors apply an advanced model, HD-MINTEQ, to model the impact of acid rain and forest harvest intensity on soil acidification, in particular base cation status.

Soil acidification is an important global issue that sometimes seems a bit drowned in the attention for soil C cycling but is very relevant indeed. As such the study is of broader interest to the community of SOIL.

The study overall seems to have been well conducted in a sound, scientific manner

and the write-up and presentation is very good. One exception is the figures: because they are effectively a multitude of figures combined in 7 larger ones they are quite small and not always immediately comprehensible. This can relatively easily be amended by reducing some of the figures, or perhaps moving some to the supplementary material.

While the study in its core is well conducted, there is however one significant issue lacking and that is linking the study to the wider context. As said, soil acidification is a global issue of great importance. However, in their combined results and discussion section, and even their conclusions, the authors limit themselves to describing the results from the studied Swedish sites only. No real attempt is made to place the results in a broader context, or even to extensively discuss them in the context of other work on soil acidification in relation to forest harvest practices. As a result the reader is left to wonder what the significance of it all is. What do the modelling results mean for soil acidification globaly? Why is this model better than existing approaches, e.g. the ProdMod and ForSAFE models mentioned in the introduction? Can the HD-MINTEQ model be applied to other forest settings around the globe? If not, what is the remaining knowledge gap to be addressed? Such questions need to be addressed, and this will mean a significant overhaul and extension of the results & discussion (perhaps better to separate both into two sections), and the conclusions.

As no new data is needed for this, this should be feasible, but it does mean major revisions, which is therefore my recommendation.
* * *

---

## Referee Comment (RC3) · Anonymous Referee #3 · 2 Oct 2018

This paper utilizes a dynamic model to assess the future impact of hypothetical timber harvest scenarios on soil chemistry at 3 sites in Sweden and compares the simulated response to the simulated impact of acidic deposition. The main findings are that timber harvesting will acidify soils (WTH slightly more than CH) but not to the extent simulated by acidic deposition. The authors also report that weathering rates will generally increase. Overall the paper is well written and the findings are what one would expect and which have been reported elsewhere – namely timber harvesting can acidify soils through base cation removal. Therefore in that sense I do not disagree with

any of the main findings and conclusions of the paper. The authors cite many of the previous papers that show essentially the same thing. However, the main question I have is whether the actual values reported are in any way meaningful. Throughout the paper the authors report changes that are relatively small (e.g. "At Aneboda, the pH across all horizons dropped by an average 0.13 (WTH) and 0.12 units (CH) compared to the NH scenario by the year 2080"). However when I look at the calibration figures and simulation figures (3-5) it is quite clear to me that parameters such as pH and base cations are very poorly matched. For example, Figure 3b shows simulated Mg in soil solution and observed values for the various horizons – they don't match very well and I don't see any clear separation of the observed data by horizon. Likewise, there is no Al3+ chemistry shown nor any description of how well the model simulations match the observed chemistry for the various soil horizons. Hence, the authors have a model that performs as expected but I have no confidence in the actual numbers nor the timeframe of the reported changes. Without a detailed evaluation of the model performance that provides the reader with some confidence that the simulated changes are in any way meaningful I cannot recommend that this paper should be published. Section 2.6 describes model calibration but looking at the figures it seems to be a very poor match for the various horizons. Other Points. 1. Quality of Figures is poor. 2. The authors show several spikes in soil solution chemistry but these cannot be validated. 3. There is no real discussion – the results/discussion section is primarily a description of the simulations and how the model is parameterized. 4. The reported weathering rate changes (see Figure 6 and 7) are so small as to be insignificant compared with other inputs/outputs (deposition/plant uptake) and of course there is no way of knowing whether this is actually happening (assumes PROFILE is correct). 5. Ignoring N dynamics is an issue – trees take up N and it can change both because of deposition and/or harvesting. 6. Conclusions – first sentence depends where in the world you are.

---

## Author Comment (AC1) · 29 Nov 2018

RE1 = referee 1 ; AU = author's response, (y) = comment no. y

RE1(1): It is not clear exactly what the purpose of the study is. Is it to demonstrate the utility of the HD-MINTEQ model for the projection of future soil chemistry under different forest management practices, to specifically make projections that can inform forest management practice or a combination of both? The manner in which the paper is written suggests largely the latter, in which case I have reservations regarding the

confidence that can be placed in the model as currently constructed as a comprehensive predictor of soil and soil solution chemistry. The lack of a specific submodel for N species transformations is in my opinion a potentially significant shortcoming.

AU(1): We have added the following sentences to the end of the introduction to clarify the objective of our paper: "The objectives of this paper are to (i) investigate possible long-term impacts of two theoretical future harvesting scenarios on the acidification and base cation status using a novel dynamic model, HD-MINTEQ, and (ii) compare biomass-induced acidification to the historical acidification that took place during the 20th century due to acid rain." We have also changed some of the language in the discussion section to deemphasize these simulations as a comprehensive predictor of soil chemistry—that was not the intent of our paper. One sentence that exemplifies our stance on this follows: "However, because the HD-MINTEQ simulations here produced stronger effects on soil Ca compared to those observed in the field (Zetterberg et al., 2016), it seems probable that the acidification effect as predicted by HD-MINTEQ may be regarded as a 'worst-case scenario', and that the real acidification, at least over the first rotation period, may be even smaller."

We would also like to point out our admitted shortcoming concerning N cycling: "However, at this point it needs to be added that these conclusions may not be relevant in cases when nitrification following harvest is substantial, in which case the acidification effect could be considerably larger; this possibility was not considered in our simulations."

RE1(2): It is not clear what specifically is new about this work in the context of modelling soil and soil solution chemistry. The model is state of the art for its description of inorganic matter equilibrium interactions and the description of BC weathering, but it falls somewhat short of the state of the art in other areas.

AU(2): The unique aspect of this work is the comparison between the acidification effect of historical S deposition, and the effect that can be expected from forest harvesting. We have tried to clarify this a little more, especially in the Conclusions section, which is partly rephrased to highlight this in a better way.

RE1(3): Given these acknowledged shortcomings of the model, it would seem appropriate to me to place more emphasis than has been placed, on assessing the ability of the model to predict the trends and/or magnitudes in soil and soil solution chemistry. The results and discussion section effectively takes the model predictions as 'correct' in its assessment of future trends.

AU(3): Thanks for pointing this out. It was certainly not our intent to push this model off as 100% correct. We have made some changes to the language to clarify that these were modeled simulations and that there are of course limitations to them. Below are some examples that will be found in the revised text: "...because the HD-MINTEQ simulations here produced stronger effects on soil Ca compared to those observed in the field (Zetterberg et al., 2016), it seems probable that the acidification effect as predicted by HD-MINTEQ may be regarded as a 'worst-case scenario', and that the real acidification, at least over the first rotation period, may be even smaller." "...at this point it needs to be added that these conclusions may not be relevant in cases when nitrification following harvest is substantial, in which case the acidification effect could be considerably larger; this possibility was not considered in our simulations. Most Swedish forests are N-limited (Högberg et al., 2017), but increased nitrate concentrations are found in soil solution for some years after final felling. Nitrification is dependent on site productivity, which is between 4 and 8 m3 ha-1 yr-1 in the sites studied. According to the estimates of Futter et al. (2010), the total accumulated harvest effect should generally not exceed 220 and 500 meq NO3- m-2 for site productivities of 4 and 8 m3 ha-1 yr-1, respectively (Futter et al., 2010), indicating rather modest nitrification effects on the long-term acid-base status of the soils. As an example, this value represents between 5 and 15 % of the atmospherically deposited BC over a full rotation period, hence nitrification is a relatively minor proton source as compared to other processes in the forest soils under study."
RE1(4): I feel that the paper would be strengthened by an assessment of the model capabilities alongside the discussion of the future projections, coupled with an assessment of where the model could be strengthened e.g. a better description of N transformations (or, of course, arguments as to why such 'improvements' would not greatly improve predictions!).

AU(4): We have tried to improve the text (i) by providing more discussion on the reasons behind the differences between model and observations, and (ii) by identifying areas where the model can be strengthened. As concerns the first of these aspects, we have provided new text towards the end of section 3.1, as follows: "For the Gårdsjön and Kindla sites, the modelled results align well with the lysimeter data, with a few exceptions. Interestingly, the model underestimated Mg2+ concentrations at both sites. This could be caused by the use of A2M estimates, i.e., that the normalization model underestimated the presence of easily weathered Mg-containing minerals. In a recent study, Casetou-Gustafson et al. (2018) compared A2M with the mineralogy obtained by X-ray powder diffraction (XRPD) for two soils that were similar to the soils studied here. They found that trioctahedral mica and hydrobiotite were consistently underestimated by A2M, which is consistent with our modelling results as these Mg-containing minerals have relatively high weathering rates. Moreover at Kindla, SO42- concentrations were underestimated. There may be several explanations, but one possibility is mineralization and oxidation of organically bound S (Löfgren et al., 2001, 2014). The delay in SO42- decrease at Kindla was, however, predicted well in the B1 horizon of the model.

For the Aneboda site, the discrepancies between model and observations were more substantial. For example, while SO42- and pH were grossly underestimated, Ca2+ and Mg2+ were overestimated. It is important to note that the lysimeter data plotted in Figures 1, 2 and 3 (previously 3, 4 and 5) are averages based on data from several lysimeters, and it has previously been observed that there are large variations in the results of individual lysimeters at the Aneboda site (Löfgren et al., 2010; Löfgren et al.,

2011; Löfgren et al., 2014). As an example, for the B horizon the averaged results are based on 8 lysimeters. Three of these, nos. 7102, 7104 and 7105, had results that were clearly divergent from the others (new Fig. S4, Supplement). Dissolved $SO_4^{2-}$, $Ca^{2+}$ and $Mg^{2+}$ were all considerably higher, whereas the pH was lower. Possible reasons include a net mineralization and oxidation of organically bound sulphur in response to decreased S deposition (Löfgren et al., 2001, 2014), a process which was not taken into account in the model. It may also be observed that if the results from the three lysimeters were removed, there would be a clearly improved agreement between the model and the observations."

Discussion of possible improvements is provided in the final part of section 3.3, as follows: "Although the HD-MINTEQ model provided simulation results that appear reasonable, a future task is to upgrade the model to include N transformations, so that the effects arising from e.g. N deposition and nitrification can be more accurately assessed than was possible in the current work. Further, improved estimates of the mineralogical composition through e.g. X-ray diffraction would be desirable to avoid the mismatch in individual base cations, as was observed for $Mg^{2+}$."

RE1(5): Finally (and returning somewhat towards my first bullet point) I cannot see any broader context - why is there a need to make these predictions in the first place? For example, can the results tell us anything about the effects of different future management practices on forest viability? I appreciate that this is somewhat beyond the scope of this paper as it relates to relationships between soil chemistry and tree growth/performance, but I do feel that some broader context is needed to emphasise the importance and usefulness of this work.

AU(5): As should now be clear from the objectives, our focus is not to present a toolbox for accurately predicting the effects of different future management practices, but rather to contribute to the discussion of these impacts by comparing different scenarios with a simplified model, by for example comparing these effects to those of historical acid rain.

RE1(6): p3 line 23. How is the supply of N species concentrations realised over the whole simulation period? It is currently unclear how exactly N species are handled. AU(6): It is held constant according to the values in Table 1. We have added the following text to clarify this: "HD-MINTEQ does not simulate N chemistry; instead dissolved $NH_4^+$ and $NO_3^-$ in the different horizons are given as input data, and are held constant (Table 1)"

RE1(7): p3 line 24. It is not clear what is meant by 'The soil pools of Al and organic C were assumed to be constant over the simulated time period. Does this mean that losses of Al and C in drainage are assumed to be completely replenished? Does the soil pool of Al mean the geochemically active pool, or is there a non-reactive component that supplies active Al by weathering?

AU(7): Yes, Al and C are assumed to be completely replenished, and the soil pool of Al does refer to the geochemically active pool. We have included the word "geochemically active" to clarify the last point, also see model description in Löfgren et al. (2017), which is cited in the paper.

RE1(8): p5 line 10. Why was ForSAFE only used at Gårdsjön?

AU(8): For the purpose of this work we have relied on parameterisations made previously by two different research groups, by Belyazid and Moldan (2009) for the Gårdsjön site and by Zetterberg et al. (2014) for the two other sites. In the latter case, ProdMod was used to estimate uptake values, as is written in the text.

RE1(9): p7, line 4. It is unclear what is meant by uneven depletion rates or how the BC uptake rates were adjusted to compensate for this. Section 2.6. My interpretation is that pH was not used in calibration? This seems unusual given the importance of pH changes as a consequence of soil management and acidification/recovery. Why was this not done?

AU(9): It was. Lysimeter pH measurements were used to calibrate the model. BC

concentrations were adjusted to achieve modeled pH values that matched empirical pH values. This has been added to the text in section 2.6: "The modeled values of pH, dissolved inorganic Al and other BC concentrations from 1993 to 2014 were then compared to soil solution data from the same period taken from various depths and binned into either horizon O, E, or B before being plotted."

RE1(10): Section 3 (Results and discussion). There appears to be no comparison of the model predictions of pH, BC and sulphate with the observations. This to me is a serious omission since the usefulness of the model in predicting the future effects of forest management depends critically on its ability to predict how the soil chemistry has responded to changing inputs and management in the past. Does the model predict the trends and/or magnitudes of the observations over time to a degree that provides sufficient confidence for it to be a useful tool for investigating the projecting future trends? If not, can potential reasons for discrepancies be suggested and means to further develop the model to address such reasons be proposed? AU(10): We have including new texts concerning the comparisons of model predictions with observations. C.f. replies and texts in response to comments 4 and 9.

RE1(11): p10, line 4 onwards. As an explanation of a modification to the model in the light of initial runs, this discussion would be better placed close to the model calibration section.

AU(11): Agreed, we have moved this text to section 2.6, immediately after the description of how the model was calibrated.

RE1(12): Figures 3-5. It is somewhat difficult to follow the graphs but my impression is that pH in B1&B2 is modelled as being consistently higher than pH in E, but that the observations suggest that pH is either quite similar in both horizons or tends to be higher in E. This is an example where the model capabilities need to be assessed against the observations - is the model a useful tool for assessing what is being projected here?

AU(12): The pH values in B1 and B2 were consistently higher than pH in E, both for the

observations and for the model. The captions were wrong! We have changed Figures 1-3 (previously 4-6) to correct this.

RE1(13): Figures 3-5. It is rather difficult to match observations and model results on those panels where observations are present. I suggest redrawing to make the connection more obvious - a simple way would be to colour the observation points according to horizon in the same way as the model lines (though the B1/B2 lines might need to be recoloured also).

AU(13): We thank the reviewer for the comment. We have changed the colors of the observed data markers to match the modeled horizon in Figures 1, 2, and 3 (previously 3, 4, and 5).

RE1(14): Figures 3-5. The symbols noted in the captions do not correspond with those on the graphs - specifically the graphs have crosses, which are not listed in the captions. This needs correction. I've assumed that crosses are supposed to be open squares.

AU(14): We have made these changes, thank you.

---

## Author Comment (AC2) · 29 Nov 2018

RE2 = referee 2 ; AU = author's response, (y) = comment no. y

RE2(1): In their manuscript, the authors apply an advanced model, HD-MINTEQ, to model the impact of acid rain and forest harvest intensity on soil acidification, in particular base cation status. Soil acidification is an important global issue that sometimes seems a bit drowned in the attention for soil C cycling but is very relevant indeed. As such the study is of broader interest to the community of SOIL. The study overall

seems to have been well conducted in a sound, scientific manner and the write-up and presentation is very good. One exception is the figures: because they are effectively a multitude of figures combined in 7 larger ones they are quite small and not always immediately comprehensible. This can relatively easily be amended by reducing some of the figures, or perhaps moving some to the supplementary material.

AU(1): We agree to reducing the number of figures in the main text, so we suggest moving Fig. 1 and Fig. 2 to the Supplement. The figures have been renumbered as a consequence. Multiple panes are required because there are 3 sites being studied, and within these sites, multiple parameters.

RE2(2): While the study in its core is well conducted, there is however one significant issue lacking and that is linking the study to the wider context. As said, soil acidification is a global issue of great importance. However, in their combined results and discussion section, and even their conclusions, the authors limit themselves to describing the results from the studied Swedish sites only. No real attempt is made to place the results in a broader context, or even to extensively discuss them in the context of other work on soil acidification in relation to forest harvest practices. As a result the reader is left to wonder what the significance of it all is. What do the modelling results mean for soil acidification globaly?

AU(2): We have included the following text towards the end of the Discussion, immediately before Conclusions: "Although the model was parameterized for three Swedish forest sites, the main trends are likely to be valid also for forest soils in other parts of the world, i.e. that forest management practices are not likely to result in strong acidification effects within one full rotation period. However, these results should not be extrapolated to longer time perspectives, as certain drivers of the model may be increasingly uncertain with time. For example, it is not known to what extent the base cation uptake behavior will differ between NH, CH and WTH scenarios over a period of several rotations."

RE2(3): Why is this model better than existing approaches, e.g. the ProdMod and For-SAFE models mentioned in the introduction? Can the HD-MINTEQ model be applied to other forest settings around the globe? If not, what is the remaining knowledge gap to be addressed? Such questions need to be addressed, and this will mean a significant overhaul and extension of the results & discussion (perhaps better to separate both into two sections), and the conclusions. As no new data is needed for this, this should be feasible, but it does mean major revisions, which is therefore my recommendation.

AU(3): ProdMod is not a dynamic acidification model that can be used for this. The advantage of using HD-MINTEQ over ForSAFE is its state-of-the-art descriptions of base cation and aluminium chemistry. This was mentioned already in the first manuscript version, but this has now been further clarified towards the end of the Introduction, in the following sentence: "An advantage of using the HD-MINTEQ model over e.g. ForSAFE and MAGIC is that the former is based on state-of-the-art descriptions of aluminium (Al) and base cation (BC) chemistry, which are probably more accurate (Gustafsson et al., 2018)." As indicated here, these aspects are more fully discussed in a companion paper. As regards the applicability of the model to other forest ecosystems, there is nothing that prevents it from being used also there. The same pros and cons (i.e. the omission of the N chemistry) will be relevant. See also the new text in response to comment no. 2.

---

## Author Comment (AC3) · 29 Nov 2018

RE3 = referee 3 ; AU = author's response, (y) = comment no. y

RE3(1): However, the main question I have is whether the actual values reported are in any way meaningful. Throughout the paper the authors report changes that are relatively small (e.g. "At Aneboda, the pH across all horizons dropped by an average 0.13 (WTH) and 0.12 units (CH) compared to the NH scenario by the year 2080"). However when I look at the calibration figures and simulation figures (3-5) it is quite

clear to me that parameters such as pH and base cations are very poorly matched. For example, Figure 3b shows simulated Mg in soil solution and observed values for the various horizons – they don't match very well and I don't see any clear separation of the observed data by horizon.

AU(1): Thank you for this comment. You are correct that we should have provided a discussion of why the model predictions did not always agree with the observations, we regret that we didn't in the first version of the manuscript. For Aneboda the results from individual lysimeters differ considerably, which is important to note when addressing the discrepancies, and therefore we have included a new supplemental figure (S4) that shows the individual lysimeter results. We have now included text towards the end of section 3.1 that addresses these issues, as follows: "For the Gårdsjön and Kindla sites, the modelled results align well with the lysimeter data, with a few exceptions. Interestingly, the model underestimated $Mg^{2+}$ concentrations at both sites. This could be caused by the use of A2M estimates, i.e., that the normalization model underestimated the presence of easily weathered Mg-containing minerals. In a recent study, Casetou-Gustafson et al. (2018) compared A2M with the mineralogy obtained by X-ray powder diffraction (XRPD) for two soils that were similar to the soils studied here. They found that trioctahedral mica and hydrobiotite were consistently underestimated by A2M, which is consistent with our modelling results as these Mg-containing minerals have relatively high weathering rates. Moreover at Kindla, $SO_4^{2-}$ concentrations were underestimated. There may be several explanations, but one possibility is mineralization and oxidation of organically bound S (Löfgren et al., 2001, 2014). The delay in $SO_4^{2-}$ decrease at Kindla was, however, predicted well in the B1 horizon of the model.

For the Aneboda site, the discrepancies between model and observations were more substantial. For example, while $SO_4^{2-}$ and pH were grossly underestimated, $Ca^{2+}$ and $Mg^{2+}$ were overestimated. It is important to note that the lysimeter data plotted in Figures 1, 2 and 3 (previously 3, 4 and 5) are averages based on data from several lysimeters, and it has previously been observed that there are large variations in the

results of individual lysimeters at the Aneboda site (Löfgren et al., 2010; Löfgren et al., 2011; Löfgren et al., 2014). As an example, for the B horizon the averaged results are based on 8 lysimeters. Three of these, nos. 7102, 7104 and 7105, had results that were clearly divergent from the others (Fig. S4, Supplement). Dissolved $SO_4^{2-}$, $Ca^{2+}$ and $Mg^{2+}$ were all considerably higher, whereas the pH was lower. Possible reasons include a net mineralization and oxidation of organically bound sulphur in response to decreased S deposition (Löfgren et al., 2001, 2014), a process which was not taken into account in the model. It may also be observed that if the results from the three lysimeters were removed, there would be a clearly improved agreement between the model and the observations."

RE3(2): Likewise, there is no $Al^{3+}$ chemistry shown nor any description of how well the model simulations match the observed chemistry for the various soil horizons.

AU(2): In this case, the reason is that there were no measured $Al^{3+}$ lysimeter data available.

RE3(3): Hence, the authors have a model that performs as expected but I have no confidence in the actual numbers nor the timeframe of the reported changes. Without a detailed evaluation of the model performance that provides the reader with some confidence that the simulated changes are in any way meaningful I cannot recommend that this paper should be published.

AU(3): As mentioned in the response to comment no. 2, we have now provided additional discussion of the model performance. Moreover, we never intended to give the impression that we thought our model was perfect. We do believe, however, that the model gives an idea of the direction of the changes, and of the relative significance of acid rain and forest harvesting, as the underlying chemical processes are well known and integrated into the model (with the exception of N). However, we have now included text that underlines the uncertainty of the model, above all in the following text in section 3.2: "The absolute magnitude of the model-predicted changes is of course

uncertain, not least in the light of the mixed success of the model to predict the available lysimeter data, as discussed in the previous section. Nevertheless, the simulated results suggest that the acidification due to a harvesting event at 2020 would be less impactful, over the time range studied, than that of historic atmospheric acidification."

RE3(4): Section 2.6 describes model calibration but looking at the figures it seems to be a very poor match for the various horizons.

AU(4): See response to comment no. 1

RE3(5): Quality of Figures is poor.

AU(5): High quality images will be uploaded.

RE3(6): The authors show several spikes in soil solution chemistry but these cannot be validated.

AU(6): The spikes are reported also in previous studies and have been mentioned in the manuscript:

Line 19, page 7: "The anomalous blip in pH (and BC) at Aneboda occurring between 2008 and 2018 is the result of a large storm that downed many trees in 2005, followed by a bark beetle infestation, resulting in acidification and increased dissolved Ca2+ (Löfgren et al., 2014). " Line 31, page 4: "The dips in deposition at Gårdsjön in the early 1900s and Kindla around 1890 were due to historical harvesting events. The dips that occur in 2020 were due to the simulated harvest scenarios."

RE3(7): There is no real discussion – the results/discussion section is primarily a description of the simulations and how the model is parameterized.

AU(7): See response to comments no. 1 and 3

RE3(8): The reported weathering rate changes (see Figure 6 and 7) are so small as to be insignificant compared with other inputs/outputs (deposition/plant uptake) and of course there is no way of knowing whether this is actually happening (assumes

PROFILE is correct).

AU(8): Thank you for bringing this up, the changes are indeed very small in comparison to other inputs and outputs. We have considered this more closely, and you are quite right that the main message of this part of the manuscript should more logically be that the model-predicted weathering rates remained more or less the same, both during the historic acidification era and during the harvest scenarios. This has prompted us to rewrite certain sections in which we discuss these results, e.g., as follows: Abstract, last lines: "In general, the predicted changes in weathering rates were very small, which can be explained by the net effect of decreased pH and increased $Al_{3+}$, which affected the weathering rate in opposite ways. Similarly, weathering rates after the harvesting scenarios in 2020 remained largely unchanged according to the model." Section 3.3: "During the historic acidification period, the annual weathering rates at several sites and horizons (Aneboda: E; Gårdsjön: E, B1, and B2; Kindla: B1 and B2) actually decreased (Fig. S5), although not by much. This is due to the brakes in the weathering function (i.e. from $Al_{3+}$ and BC, Eqs. 2 and 3) and to the fact that total dissolved Al and BC were much higher during this period (Figs. 1, 2 and 3). In other words, the increased weathering rate expected from a decreased pH was offset by the increase in dissolved Al and BC, resulting in a very small net effect, which according to PROFILE was negative. However, the exact patterns varied from site to site, and from layer to layer. The low weathering rates in the E horizon at Gårdsjön were likely due to its mineralogy, i.e., minerals other than quartz, K-feldspar, and plagioclase were essentially absent from the E horizon, but they were present further down in the profile. Another factor leading to low weathering rates in the Gårdsjön E horizon was the relatively thin layer thickness.

Contrary to the decrease in weathering rates during the historic acidification of the 1970s, the simulated weathering rates after the harvesting scenarios at 2020 generally increased compared to NH by 2080 (Fig. 5) although again, the net change was very small. However, the dynamics were quite different across each layer and site. The B1

horizon at Aneboda saw the strongest increase in weathering rates after harvesting, increasing by 9% (CH) and 11% (WTH) by 2080 compared to the NH scenario. The other horizons at Aneboda (E and B2) had almost unchanged weathering rates (<2% increase by 2080 compared to a NH scenario). It is worth noting that the difference between harvesting events and NH at the Aneboda B1 and B2 horizons would likely continue to diverge beyond 2080. At Aneboda, by the year 2080, the sum of weathered BC for the CH and WTH scenarios were 32 (CH) and 46 (WTH) meq m-2 higher, respectively, than the BC weathered in the NH scenario (Fig. S6). To put this into perspective, this difference is equivalent to only 1.1 % and 1.7 %, respectively, of the atmospherically deposited BC in the WTH plots over the same period. Such a small change in the weathering rate cannot be experimentally verified, and is unlikely to be of any ecological significance."

Consequential changes were made also in the remaining part of section 3.3, when discussing the results of Gårdsjön and Kindla, and also in the Conclusions.

RE3(9): Ignoring N dynamics is an issue – trees take up N and it can change both because of deposition and/or harvesting.

AU(9) To what extent this is an issue depends largely on to what extent nitrification (and subsequent nitrate release) is occurring. We acknowledged this already in the first version of the manuscript, but to make it clearer we have also added more text discussing this. Here are some examples:

Section 2.6, immediately before Results and Discussion: "However, at this point it needs to be added that these conclusions may not be relevant in cases when nitrification following harvest is substantial, in which case the acidification effect could be considerably larger; this possibility was not considered in our simulations. Most Swedish forests are N-limited (Högberg et al., 2017), but increased nitrate concentrations are found in soil solution for some years after final felling. Nitrification is dependent on site productivity, which is between 4 and 8 m3 ha-1 yr-1 in the sites studied. According

to the estimates of Futter et al. (2010), the total accumulated harvest effect should generally not exceed 220 and 500 meq NO3- m-2 for site productivities of 4 and 8 m3 ha-1 yr-1, respectively (Futter et al., 2010), indicating rather modest nitrification effects on the long-term acid-base status of the soils. As an example, this value represents between 5 and 15 % of the atmospherically deposited BC over a full rotation period, hence nitrification is a relatively minor proton source as compared to other processes in the forest soils under study."

Page 13, lines 15-18: "As mentioned previously, limitations in the model prevented us from addressing possible nitrification effects resulting from long-term N deposition, which may influence these results. A future task is to upgrade the HD-MINTEQ model to include N transformations, so that the effects arising from e.g. N deposition and nitrification can be more accurately assessed."

RE3(10): Conclusions – first sentence depends where in the world you are.

AU(10): This sentence has been removed, as its statement was debatable and in addition not necessary for the paper.